# Clinical Consequences of Incidental Durotomy during Full-Endoscopic Lumbar Decompression Surgery in Relation to Intraoperative Epidural Pressure Measurements

**DOI:** 10.3390/jpm13030381

**Published:** 2023-02-22

**Authors:** Roth A. A. Vargas, Marco Moscatelli, Marcos Vaz de Lima, Jorge Felipe Ramírez León, Morgan P. Lorio, Rossano Kepler Alvim Fiorelli, Albert E. Telfeian, John Fiallos, Ernest Braxton, Michael Song, Kai-Uwe Lewandrowski

**Affiliations:** 1RIWO Spine Center of Excellence, Department of Neurosurgery, Foundation Hospital Centro Médico Campinas, Campinas 13101-627, SP, Brazil; 2Clinica NeuroLife, Natal 59054-630, RN, Brazil; 3Department of Orthopedics and Traumatology, Santa Casa de São Paulo, “Pavilhão Fernandinho Simonsen”, São Paulo 05014-901, SP, Brazil; 4Minimally Invasive Spine Center, Bogotá, D.C., Colombia, Reina Sofía Clinic, Bogotá, D.C., Colombia, Department of Orthopaedics, Fundación Universitaria Sanitas, Bogotá 104-76, D.C., Colombia; 5Advanced Orthopedics, 499 E. Central Pkwy, Ste. 130, Altamonte Springs, FL 32701, USA; 6Department of General and Specialized Surgery, Gaffrée e Guinle University Hospital, Federal University of the State of Rio de Janeiro (UNIRIO), Rio de Janeiro 20270-004, RJ, Brazil; 7Department of Neurosurgery, Rhode Island Hospital, The Warren Alpert Medical School of Brown University, Providence, RI 02903, USA; 8Minimally Invasive Spine Center, Bogotá 104-76, D.C., Colombia; 9Vail Summit Orthopaedics & Neurosurgery, Frisco, CO 80443, USA; 10Advanced Neurosurgery, Reno, NV 89511, USA; 11Center for Advanced Spine Care of Southern Arizona, Tucson, AZ 85712, USA; 12Orthopaedic Surgery, Department of Orthopaedics, Fundación Universitaria Sanitas, Bogotá 104-76, D.C., Colombia; 13Department of Orthopedics at Hospital Universitário Gaffre e Guinle, Universidade Federal do Estado do Rio de Janeiro, Rio de Janeiro 20270-004, RJ, Brazil

**Keywords:** epidural pressure, interlaminar lumbar endoscopy, neurological complication

## Abstract

**Background**: Seizures, neurological deficits, bradycardia, and, in the worst cases, cardiac arrest may occur following incidental durotomy during routine lumbar endoscopy. Therefore, we set out to measure the intraoperative epidural pressure during lumbar endoscopic decompression surgery. **Methods**: We conducted a retrospective observational cohort study to obtain intraoperative epidural measurements with an epidural catheter-pressure transducer assembly through the spinal endoscope on 15 patients who underwent lumbar endoscopic decompression of symptomatic lumbar herniated discs and spinal stenosis. The endoscopic interlaminar technique was employed. **Results**: There were six (40.0%) female and nine (60.0%) male patients aged 49.0667 ± 11.31034, ranging from 36 to 72 years, with an average follow-up of 35.15 ± 12.48 months. Three of the fifteen patients had seizures with durotomy and one of these three had intracranial air on their postoperative brain CT. Another patient developed spinal headaches and diplopia on postoperative day one when her deteriorating neurological function was investigated with a brain computed tomography (CT) scan, showing an intraventricular hemorrhage consistent with a Fisher Grade IV subarachnoid hemorrhage. A CT angiogram did not show any abnormalities. Pressure recordings in the epidural space in nine patients ranged from 20 to 29 mm Hg with a mean of 24.33 mm Hg. **Conclusion**: Most incidental durotomies encountered during lumbar interlaminar endoscopy can be managed without formal repair and supportive care measures. The intradural spread of irrigation fluid and intraoperatively used drugs and air entrapment through an unrecognized durotomy should be suspected if patients deteriorate in the recovery room. Ascending paralysis may cause nausea, vomiting, upper and lower motor neuron symptoms, cranial nerve palsies, hypotension, bradycardia, and respiratory and cardiac arrest. The recovery team should be prepared to manage these complications.

## 1. Introduction

Endoscopic spine surgery is much more widely practiced than just five years ago [1,2,3]. The interest of patients and surgeons alike in the procedure has risen substantially, leading to a volume increase across the board [4]. Common clinical problems that are now being reliably treated with endoscopic surgery include not just lumbar herniated discs but also, to a higher degree, bony and soft tissue stenosis [5,6,7,8,9]. Stenotic lesions in the central and lateral spinal canal and foraminal stenosis can be effectively treated in the lumbar [10,11,12,13], thoracic [14,15], and cervical [16,17,18] spine with various endoscopic surgeries carried out through the transforaminal or interlaminar approach or variations thereof. Many surgeons have learned how to effectively use power drills, burrs, and specialized rongeurs to perform wide bony and soft tissue decompression, which plays a central role in interlaminar and in some uniportal/biportal (UBE) techniques [19]. As a result, previously difficult-to-reach extraforaminal disc herniations and other pain generators within the intervertebral disc and the facet joint complex can be directly visualized and treated during endoscopic decompression surgery [20,21].

With the surgical volume rising, publications on the incidence and management of complications of endoscopic lumbar decompression procedures have appeared in print, alerting spinal surgeons interested in implementing these innovative minimally invasive surgeries into their program to their potential pitfalls [12,19,22,23,24]. While complication rates are generally lower than with traditional translaminar surgeries [24], interlaminar and transforaminal endoscopic decompression surgeries are associated with their specific sets of difficulties and postoperative sequelae that could catch the novice surgeon unfamiliar with these pitfalls off guard. One such example relates to small dural tears that can easily be encountered in the posterior dural sac during the interlaminar approach or in the axilla formed between the exiting and traversing nerve roots during the transforaminal approach [22]. The latter is referred to as the hidden zone of Macnab because durotomies in this difficult-to-access and difficult-to-visualize area are most common and by far less common in any other portion of the epidural space.

A recent survey study among 97 spine surgeons suggested that durotomy rates are much higher among less experienced surgeons (1% to 10%) than the 0.1% rate previously reported by expert surgeons with thousands of cases of clinical experience. In that same study, some 70% of durotomies were reported by 20.4% of participating surgeons, and only 29% of surgeons had more than 10–15 years of clinical work experience [22]. In contrast, nearly half of the participating surgeons (49%) had only 1 to 5 years of experience with spinal endoscopy [22]. These data suggest that incidental durotomies are much more common than reported and that many endoscopic spine surgeons may not be aware of them [25]. This iatrogenic complication may be inferred in the recovery room if the patient complains of spinal headaches, neck pain, nausea, vomiting, and, in some severe cases, even vision disturbances [23]. While these symptoms may be due to a CSF leak and an associated drop in intracranial pressure, another explanation is possible. Spinal endoscopic surgery is performed with irrigation fluid, and some spine surgeons use irrigation pumps. Some may not be explicitly designed for the small compartment defined by the surgically exposed lateral recess or Kambin’s triangle [26,27]. Many ambulatory surgery centers (ASCs) use their existing equipment for endoscopic spinal surgery and, thus, generate considerable pressure and flow variations with irrigation pumps designed for knee and shoulder arthroscopies. The standard settings with these arthroscopic pumps were originally designed for much larger surgical compartments. Pressures may surge rapidly. The net result may be the dumping of large fluid volumes into the spinal canal. In the worst cases, seizures or cardiac arrest may ensue, mainly if the endoscopy was performed under local anesthesia [28,29,30], where the anesthetic medication could enter the intradural space and be carried into the brain via the endoscopic irrigation fluid.

This study aims to raise awareness of these durotomy-related complications, which likely occur at a higher rate than recognized by inexperienced endoscopic spine surgeons. The authors had three cases of postoperative seizures, transitory neurological deficits, and intracranial air entrapment, which motivated this study, in which the authors present their intraoperative epidural pressure measurements encountered during routine interlaminar endoscopies to make the case that these lesions should be suspected if an unexplained drop in blood pressure, seizures, or even cardiac arrest is encountered in the recovery room.

## 2. Materials and Methods

### 2.1. Study Design

Patients were enrolled into this retrospective observational sequential cohort study between March 2014 and April 2021 they had interlaminar decompression for herniated disc and spinal stenosis. There were 6 (40.0%) female and 9 (60.0%) male patients with an average age of 49.0667 ± 11.31034, ranging from 36 to 72 years. The average follow-up was 35.15 ± 12.48 months.

### 2.2. Patient Selection and Inclusion/Exclusion Criteria

Patients were included in the study if they presented with continuous cervical or lumbar radicular stemming from a herniated disc or spinal stenosis in the foramina or lateral spinal canal. Symptoms were required to be present for at least six months or more before the consultation, despite a recommended three-month physiotherapy regimen and medical and interventional pain management. Patients were further evaluated for clinical history and physical examination findings consistent with a disc herniation, including sensory deficits, motor weakness, and the presence of any upper and lower motor neuron signs. Necessary diagnostic imaging included standing lumbar X-rays, magnetic resonance imaging (MRI), or computed tomography (CT) scans. Patients were excluded if they had any underlying neurological condition or spinal-cord-compression-related upper motor neuron symptoms affecting their gait cycle and locomotion. Additional exclusion criteria for endoscopic decompression were severe bony central canal or excessive foraminal stenosis, excessive coronal and sagittal plane deformities above 40 degrees, conus medularis syndrome, systemic neuropathy or spinal tumors, blood dyscrasia, pregnancy, allergies, mental handicaps, or psychiatric conditions precluding adequate communication or language problems.

### 2.3. Endoscopic Surgery Technique

The authors chose an interlaminar approach to cervical and lumbar disc herniation and any associated soft tissue and bony stenosis in the lateral recess [31]. The patient is positioned (prone) and prepped in standard surgical fashion for surgery. Anatomic landmarks in the cervical and lumbar spine, such as the midline, the interlaminar window, the intervertebral disc space, and the facet joints, are marked on intraoperative PA and lateral views. Once the attack angles and the entry points are established, considering the location, size, and relationship to other anatomical structures, a small stab incision is made, typically 2 to 3 cm from the midline, aiming for the facet joint. The authors employed the RIWO VERTEBRIS™ lumbar instrument using a 30-degree optic exploiting the posterior interlaminar window. Dilators were guided directly to the ligamentum flavum using minimal fluoroscopy without a needle. After placing the working sleeve over the dilator, the operation was performed through a high-definition spine endoscope with a 30-degree viewing angle utilizing continuous gravity-assisted irrigation (Figure 1).

The system’s design was optimized to allow the complete functionality of power instruments with exceptional visualization, allowing the authors to readily recognize even the smallest durotomies. The endoscopic instruments and power burrs used had a characteristic narrow profile that minimizes trauma to the soft tissues, ligamentum flavum, and neural structures. The surgery entails enlarging the interlaminar window with a power burr by performing a laminotomy at the trailing edge of the rostral lamina. In addition, partial resection of the inferior articular process (IAP) is typically required to access the ligamentum flavum. The latter must be opened with a blunt dissector and subsequently removed using endoscopic forceps and a Kerrison rongeur. The neural structures are retracted, and the disc herniation and any other bony or soft tissue stenoses are removed. A radiofrequency probe can be used for hemostasis and any frayed soft tissue shrinking. After completing the decompression, the cannula will be turned to visualize the neural structures and control the decompression by a palpation hook. All surgeries were performed by highly experienced endoscopic spine surgeons.

### 2.4. Intraoperative Epidural Pressure Measurements

For the intraoperative epidural hydrostatic pressure measurements at the endoscopic decompression site, a 1.8 mm diameter transparent catheter with three lateral orifices was guided down the endoscopic working channel and held right next to the surgical instrument used to decompress the herniated disc. This epidural/arterial catheter was attached to an Abbott Medex Disposable IBP Transducer (#IP-MX-300), plugged into a Dräger—the Perseus A500 anesthesia machine with a monitor. After the completion of decompression, the catheter was positioned close to the dural sac or right next to the incidental dural tear if one was encountered (Figure 2).

Epidural hydrostatic pressures were taken at least three times in the same decompression area with and without concomitant use of the radiofrequency probe to see whether temperature variations caused by the probe would cause hydrostatic pressure changes in the epidural space. Pressure measurements were printed out and manually transferred into an excel database from which they were averaged and analyzed statistically as described below. Our epidural hydrostatic pressure measurements have their foundations are in prior work [32].

### 2.5. Statistical and Outcome Analysis

Postoperatively, patients were followed during the first 6–12 weeks after their surgery to monitor their recovery and manage any postoperative problems. At the final follow-up, clinical outcomes were assessed using the visual analog pain score (VAS) for arm- or leg pain [33]. Descriptive statistics were calculated using IBM SPSS Statistics software, Version 27.0. The means, ranges, standard deviations, and percentages of all nominal variables were calculated.

## 3. Results

Our series included three patients with seizures who also had a recognized durotomy. One of these three patients had intracranial air on his postoperative brain CT after lumbar endoscopy (Figure 3). Another one suffered from cardiac arrest.

The remaining patient of these three developed spinal headaches and diplopia on postoperative day one when her deteriorating neurological function was investigated with a brain computed tomography (CT) scan showing intraventricular hemorrhage consistent with a Fisher Grade IV subarachnoid hemorrhage [34] that did however, ultimately not demonstrate a cerebral aneurysm or AV-malformations after a surveillance CT angiogram was performed at three months postoperatively. The seizures in all three patients were treated medically and the patients were stabilized. 

The most common surgical indication was unrelenting pain from a herniated nucleus pulposus (HNP) and lateral recess stenosis. Interlaminar endoscopic decompressions were exclusively performed in all 14 of the 15 study patients at the L4/5 and L5/S1 levels. The remaining patient in our series underwent interlaminar C6/7 endoscopic discectomy surgery. All patients underwent single-level unilateral operations. The demographic and clinical outcome data are listed in Table 1. The mean preoperative VAS was 6.6667 ± 1.54303, and the mean postoperative VAS was 1.4000 ± 1.12122. The mean VAS reduction was 5.26667 ± 1.27988, which was a statistically significant reduction based on paired *t*-testing with a 95% confidence interval of a difference lower limit of 4.55789 and an upper limit of 5.97544 (t = 15.937; *p* < 0.0001). The intraoperative pressure recordings in the epidural space were averaged for each patient and ranged from 20 to 29 mm Hg. The mean epidural pressure for the entire patient series was 24.33 ± 3.13 mm Hg. Sealing the endoscope’s working channel did not substantially raise the recorded epidural pressure. It never exceeded 30 mg Hg (Figure 4).

## 4. Discussion

Endoscopic spine surgery is gaining popularity for structures beyond the lumbar spine. The worldwide publishing activity reflects an increasing surgical volume. While the preference for a particular surgical approach may vary from surgeon to surgeon, a recent survey indicated that transforaminal and interlaminar endoscopic approaches to the lumbar spine are among the most popular techniques [3]. The transforaminal approach and technique are the most suitable for extraforaminal herniations and central compressive pathologies underneath the dural sac [35]. The latter pathology is much harder to access using the interlaminar approach, ideal for stenosis in the lateral recess and posterior central canal [36]. Incidental durotomies may be encountered during either of these techniques [37]. Although a recent survey among endoscopic spine surgeons revealed that the interlaminar endoscopic decompression technique might have a slightly higher incidence of dural tears than the transforaminal technique application [22], both methods are prone to dural sac injury, mainly if power drills and burrs are used [38]. These power instruments have significantly broadened the surgical indications that can be effectively treated with spinal endoscopy, from a simple herniated disc to the more complex bony and soft tissue stenosis often encountered in elderly patients. Therefore, the associated durotomies are predominantly found in the axilla between the exiting and traversing nerve roots with the transforaminal technique and in the posterior and lateral dural sacs with the interlaminar technique [22,37,38,39,40,41,42,43].

As illustrated by our small cohort study, the actual rate of coincidental durotomies may be higher than reported in the literature. Therefore, the authors wanted to call attention to some of the graver clinical complications observed during their long-standing carrier as endoscopic spine surgeons. Many of this article’s authors have performed endoscopic spinal decompression surgery for more than 15 years and have encountered several durotomies. The literature suggests that the durotomy rate across the board is about 1% [23,37,38,39,40,41,42,43], although some authors reported incidence rates one magnitude lower in skilled hands [22]. There is less consensus on how to manage them [22]. While a recent survey suggested that many durotomy patients do not require aggressive management and can be treated conservatively with bed rest and rehydration [22], this trivialization of durotomies may also lead to the delayed management of postoperative complications such as diplopia, headaches, nerve palsies, seizures, hypotension, or, in the worst cases, respiratory and cardiac arrest if the postoperative recovery team does not understand the root cause.

In this article, the number one item on the authors’ agenda is to raise awareness for incidental durotomies during routine spinal endoscopy and alert the endoscopic spine surgeon to some of the more far-reaching consequences that could destabilize the patient’s neurological, respiratory, or cardiovascular system. Our series had three patients with severe reactions to the incidental durotomy: all had seizures one of which had free air in the brain, one suffered cardiac arrest, and another developed diplopia and had blood in the brain similar to a Fisher grade IV subarachnoid hemorrhage. The unintended consequences of overly aggressive endoscopic decompression surgery and the delayed institution of medical and supportive care measures could be far-reaching. Therefore, the authors were motivated to measure the typical epidural pressures during routine lumbar interlaminar endoscopic decompression surgery when instruments are introduced, removed, and reintroduced constantly. Since the intraoperatively measured irrigation pressure ranged between 20 to 29 mm Hg with an average of 24.33 mm Hg and is consequently higher than the intradural and intravenous pressure typically reported at 5 – 15 mg Hg; it seems reasonable to assume that irrigation fluid – in most cases 0.9% normal saline – and blood may enter the patient’s intradural space and travel via the neuroaxis into the brain’s subarachnoid space and ventricles. From the durotomy site, it can spread and flow rostrally where it can cause hydrocephalus. 

A recent spinal endoscopy study investigated the cervical epidural pressure (CEP) in 20 patients undergoing single-level biportal endoscopic lumbar discectomy (BELD) [44]. The authors noted an insignificant increase from baseline (*p* = 0.24) with a 17.3 ± 8.62 mm Hg mean CEP when establishing the access portal and working space. However, the mean CEP increased to 35.1 ± 11.44 mm Hg when the epidural space was connected to the outer working space. The maximum recorded CEP was 45.2 ± 21.92 mm Hg seconds after opening the epidural space [44]. These pressure numbers are higher than what we observed and could be explained by the use of an irrigation pump system in that study with a maximum pressure setting of 80 mm Hg. However, the authors had no dural tears or patients with neurological problems suggesting that irrigation and temporary elevations in the epidural pressure without durotomy are not necessarily problematic. More recent research indicates that the size of the durotomy may also play a role. Incidental durotomies obtained during transforaminal surgery are often 1 mm puncture holes versus those encountered during the interlaminar approach being on average 1 cm in length and width [22]. The authors’ series exclusively employed the interlaminar approach suggesting the complications reported herein are more likely to occur with larger dural tears than with more minor ones. However, this relationship should be formally investigated.

In the cervical durotomy patient, the intradural spread of blood and irrigation fluid caused the intensivist and neurosurgical team to initially think of an aneurysmal bleeding or an AV angioma since it resembled its classic appearance of a Fischer grade IV subarachnoid bleeding on computed tomography (CT) scanning of the brain [34,45]. They thought they had to manage intracranial bleeding or aneurysm. Air entrapment in the venous or neuraxial system also seems possible since the air is frequently introduced into the spinal canal via the endoscope when surgical instruments and power tools are deployed through the endoscope’s central working channel. These complications are thankfully infrequent but have been reported. A case of abducens paresis has been reported following a dural tear with subsequent fistula formation during a thoracic discectomy surgery [46]. The author’s review of the literature identified seven more cases. He concluded that the abducens and the vagal nerve, the longest cranial nerve are the most sensitive to intracranial pressure variations. The vagal nerve’s innervation of the heart explains why a durotomy-induced intradural spread during irrigated spinal endoscopy with possibly higher intracranial pressures may impact heart function and lead to bradycardia and, in the worst cases, cardiac arrest. Brain sagging may be another likely mechanisms involved in these dreadful complications. In- and outflow of irrigation fluid from the dural sac is conceivable, mainly when the surgeon introduces instruments and attempts to increase irrigation flow and pressure by sealing the endoscope’s working channel, applies suction, or adjusts the setting at the irrigation pump. Our measurements corroborate these stipulations where the epidural pressure rose to under 30 mm Hg if the surgeon sealed the end of the endoscope’s working channel. This maneuver is a common trick used by the experienced endoscopist to manage flow and pressure when there is marginal visualization due to bleeding or debris.

The type of anesthesia employed during spinal endoscopy also deserves some discussion. Many surgeons operate on patients under local anesthesia and sedation. Others perform it on patients under general anesthesia and use local anesthesia of the surgical tract and foramen to reduce intraoperative stimulation. Others perform the endoscopic operation on patients under epidural anesthesia in the elderly with COPD and other medical comorbidities. The intradural spread of the commonly used local anesthetics bupivacaine and lidocaine effectively causes ascending paralysis from spinal anesthesia. Complications from ascending paralysis include but are not limited to nausea, vomiting, upper and lower motor neuron symptoms, cranial nerve palsies, hypotension, bradycardia, and respiratory and cardiac arrest. While a discussion of the most appropriate type of anesthesia to use for lumbar endoscopy may be beyond the scope of this article, surgeons should consider an unrecognized durotomy with an intradural spread of fluid, air, and intraoperatively applied drugs if their patient displays some of the above-described symptoms that seem out of proportion with the type of operation performed. A limited work-up with a head CT to rule out other intracranial pathologies may be indicated. Most patients will recover uneventfully with supportive care measures. Our study is limited by its observational nature and small patient number. However, the few patients with serious complications we treated prompted us to perform the epidural pressure measurements reported herein.

## 5. Conclusions

The authors’ study illustrated that the pressure associated with interlaminar irrigated lumbar endoscopy measured at the epidural decompression site is relatively low, averaging 24.33 mm Hg. However, it is above the commonly found intradural pressure of 5 to 15 mm Hg [47]. Incidental durotomies may cause unintended intradural flow scenarios where irrigation fluid, blood, air, or drugs may spread along the neuroaxis. While most incidental durotomies can be managed without a formal repair, patients may develop symptoms postoperative recovery that seem out of proportion in severity with those ordinarily observed. Surgeons should consider unrecognized durotomy and manage the patient with those considerations in mind. Aggressive management is only reserved for those patients who develop postoperative seizures or cardiac arrest.

## Figures and Tables

**Figure 1 jpm-13-00381-f001:**
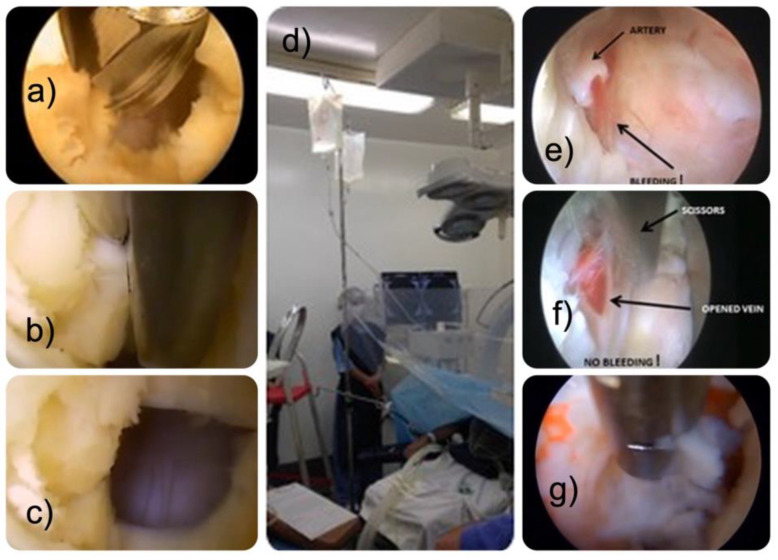
During the interlaminar approach, a power burr is used to remove part of the trailing edge of the rostral lamina (**a**), the ligamentum flavum is incised with forceps (**b**), and an opening is created (**c**). The authors prefer using gravity-assisted irrigation rather than an irrigation pump to keep the irrigation pressure relatively constant. The fluid bags are hung at 3 meters above the patient according to Ruetten’s technique (**d**). During the soft tissue decompression, arterial (**e**) and venous vessels (**f**) may have to be divided before removing the free fragments (**g**). Arterial bleeding should be controlled with a radiofrequency probe. Unrecognized durotomies are often located in the axilla between the exiting and traversing nerve root and on the undersurface of the dural sac directly in contact with the intervertebral disc. Durotomies of the posterior dural sac are common with the interlaminar approach technique. The neural elements should be carefully inspected if there is suspicion of an incidental durotomy.

**Figure 2 jpm-13-00381-f002:**
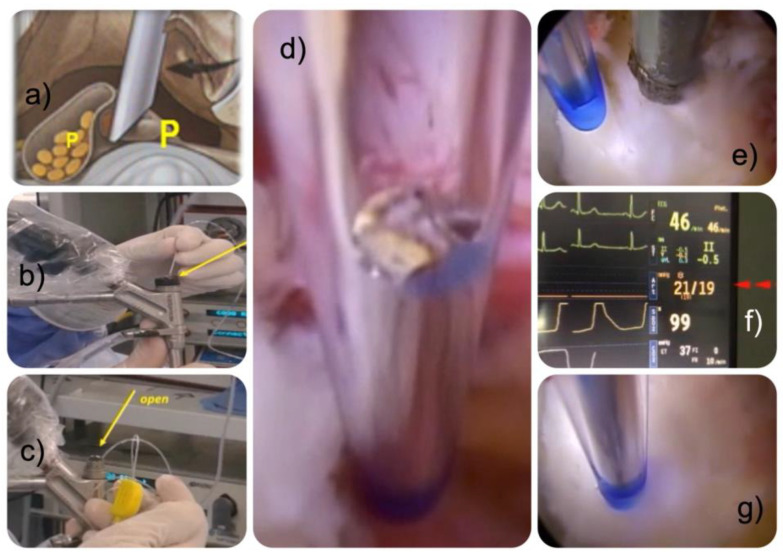
The authors’ preferred interlaminar technique is graphically depicted (**a**). The working cannula is placed into the lateral recess after a partial laminotomy and foraminotomy with some resection of the IAP. The working cannula’s bevel retracted the nerve root and protected the dural sac from injury. The intraoperative hydrostatic pressure measurements were done using a catheter probe attached to a pressure transducer (**b**). A rubber plug was used to seal the endoscope’s working channel and avoid irrigation fluid leakage and false epidural pressure readings (**c**). The catheter had one front and two side openings (**d**). It was placed into the decompression site right next to the dural sac (**e**), pressure readings (**f**) were done with (**e**), and without the simultaneous use of a radiofrequency probe (**g**) to see whether the heat associated with its use caused any pressure variations. Intraoperatively, pressure readings obtained from the epidural catheter were displayed on the anesthesia machine’s monitor and recorded. The yellow arrows indicate the indicate where the epdirual pressure measurement catheter was inserted through the working channel of the endoscope.

**Figure 3 jpm-13-00381-f003:**
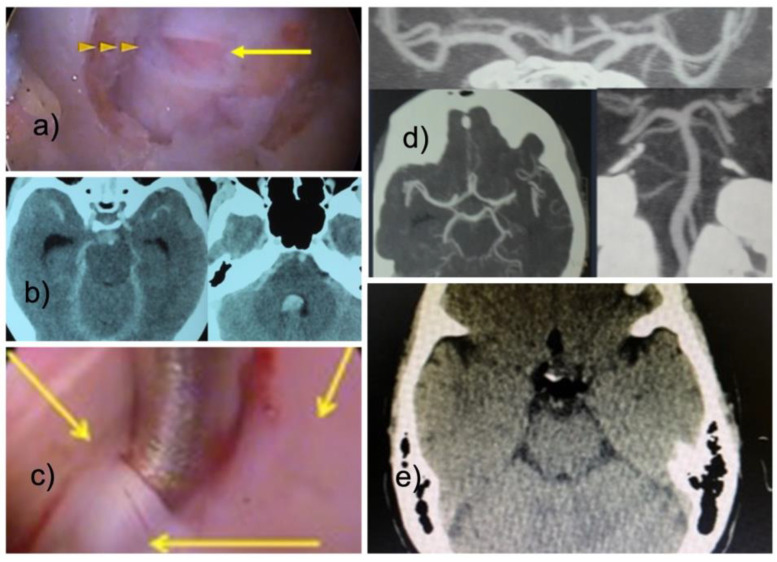
With the interlaminar endoscopic technique, incidental durotomies are frequently encountered in the posterior dural sac (**a**). They are associated with the use of power tools but can also occur during the sharp opening of the ligamentum flavum. In this exemplary patient—a 52-year-old female with a C6/7 disc herniation decompressed with the interlaminar technique - fluid entered the intradural space through the small incidental durotomy shown in panel (**a**). The surgeon had difficulty controlling the epidural bleeding encountered during the otherwise seemingly uneventful operation. He increased the hydrostatic pressure by raising and compressing the fluid bags with a Level I device to improve visualization. Postoperatively, the patient complained of diplopia and headaches. A head CT revealed subarachnoid blood and hydrocephalus typical of Fischer Grade 4 subarachnoid bleeding (**b**). A durotomy was encountered (**c**). A postoperative surveillance CT angiogram was normal and did not show any aneurysm or AV-malformations (**d**). The patient was treated with supportive care measures and recovered fully. The postoperative head CT of another lumbar durotomy patient revealed some free air (**e**).

**Figure 4 jpm-13-00381-f004:**
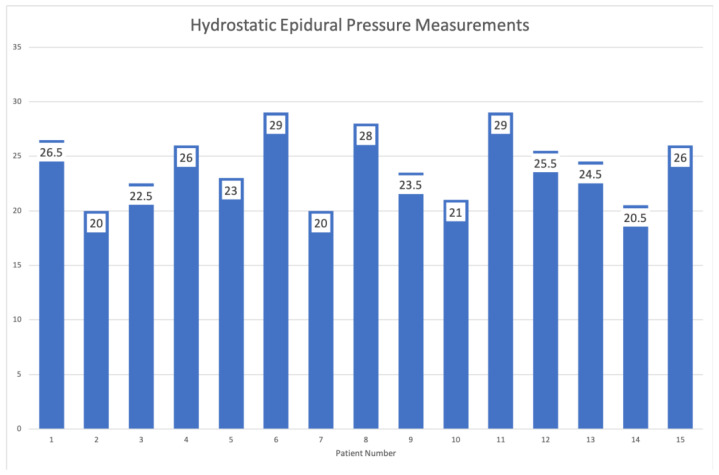
Shown are the intraoperative hydrostatic pressure recordings in the epidural space which ranged from 20 to 29 mm Hg never exceeding 30 mm Hg. The mean epidural pressure for the entire patient series was 24.33 ± 3.13 mm Hg. Concomitant radiofrequency use did not alter the measurements.

**Table 1 jpm-13-00381-t001:** Clinical outcomes in patients who underwent routine L4/5 and L5/S1 interlaminar full-endoscopic discectomies.

Patient #	Gender	Age	Surgical Level	Preop VAS Back Pain	Postop VAS Back Pain
1	M	54	L4/5	8	2
2	M	62	L4/5	6	1
3	F	46	L4/5	7	1
4	M	65	L5/S1	8	3
5	M	72	L4/5	5	0
6	F	58	L5/S1	5	1
7	F	51	L5/S1	9	2
8	F	46	L4/5	6	0
9	M	39	L5/S1	6	0
10	M	40	L4/5	4	1
11	M	36	L4/5	9	1
12	M	39	L5/S1	7	2
13	F	40	L4/5	8	4
14	M	36	L4/5	5	1
15	F	52	C6/7	7	2

## Data Availability

The data presented in this study are available on request from the corresponding author.

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
