# Peer review of "Clinical Consequences of Incidental Durotomy during Full-Endoscopic Lumbar Decompression Surgery in Relation to Intraoperative Epidural Pressure Measurements"

_jpm, 2023, doi:10.3390/jpm13030381_

Round 1

Reviewer 1 Report

The study proposes an interesting topic, useful in clinical practice especially for less experienced surgeons.

The manuscript is overall well written. The abstract is comprehensive and exhaustive. The introduction is well written and clearly introduces the aim of the study. All references are correctly reported.

The methodological approach is correct. Sub-sections are widely and carefully described. Inclusion ed exclusion criteria adequately describe the population included in the study. Images and descriptions are clear and comprehensive.

Point 2.3, “surgical technique”, is well described. Authors, if they consider it appropriate, should specify whether the procedure was performed by experienced surgeons or not. According to the authors, could this technique be a dependent operator?

Results should be improved: authors might better describe clinical outcome of the population analyzed, and furthermore they might resume all data in summary charts.

Discussion is overall well written.

Conclusions are in line with the results obtained and offer a starting point for further studies.

Author Response

All changes that were made in response to the reviewer’s minor revision request are highlighted in yellow. We added outcome and demographic data tables, and one additional figure listing the hydrostatic epidural pressure measurements. We stated as requested, that all surgeries were performed by highly experienced endoscopic spine surgeons.

Reviewer 2 Report

Dear Authors,

I read your manuscript with the interest and appreciation. Below some remarks I hope will improve the manuscript. 

1.     Line 221-222 – Statistical and outcome measurements. You mentioned that you used VAS for pain and functional assessment during first 6-12 weeks and at final follow up. Unfortunately, I did find any results of VAS neither in results chapter, nor in the abstract. 

2.     In my opinion VAS is a good subjective tool to measure pain, but not to evaluate functional assessment. Among many tools used for functional assessment you could use: Oswestry Disability Index or Spinal Stenosis Symptoms Questionnaire. 

Author Response

We appreciate this reviewer's comment. We corrected the language Line 221-222 regarding outcomes as follows:

"Postoperatively, patients were followed during the first 6 – 12 weeks following their surgery to monitor their recovery and manage any postoperative problems. At final follow-up, clinical outcomes were assessed using the visual analogue pain score (VAS) for leg pain."

As indicated in the response to reviewer #1, we added a table listing the demographic and clinical outcome data and another figure visualizing the epidural pressure measurements.

We hope that this reviewers comments are satisfied. We would be glad to answer any additional comments.

Round 2

Reviewer 2 Report

Dear Authors,

I would like to thank you for the changes in the manuscript.

In my opinion your paper is ready for publication.